# Association of Glycemic Index, Glycemic Load, and Carbohydrate Intake with Antral Follicle Counts Among Subfertile Females

**DOI:** 10.3390/nu17030382

**Published:** 2025-01-21

**Authors:** Makiko Mitsunami, Maryam Kazemi, Amy R. Nichols, Lidia Mínguez-Alarcón, Victoria W. Fitz, Irene Souter, Russ Hauser, Jorge E. Chavarro

**Affiliations:** 1Department of Nutrition, Harvard T.H. Chan School of Public Health, 677 Huntington Ave, Boston, MA 02446, USA; 2Department of Environmental Health, Harvard T.H. Chan School of Public Health, 677 Huntington Ave, Boston, MA 02446, USA; 3Department of Nutrition, University of California, 1 Shields Ave, Davis, CA 95616, USA; 4Channing Division of Network Medicine, Department of Medicine, Brigham and Women’s Hospital and Harvard Medical School, 181 Longwood Avenue, Boston, MA 02115, USA; 5Fertility Center, Vincent Department of Obstetrics and Gynecology, Massachusetts General Hospital and Harvard Medical School, 32 Fruit Street, Boston, MA 02114, USA; 6Department of Epidemiology, Harvard T.H. Chan School of Public Health, 677 Huntington Ave, Boston, MA 02446, USA

**Keywords:** glycemic index, glycemic load, carbohydrate quality, antral follicle count, ovarian reserve

## Abstract

Background/Objectives: Few studies have investigated the association of dietary glycemic index (GI), glycemic load (GL), and carbohydrate intake with antral follicle count (AFC). This study aimed to investigate the association of total carbohydrate intake and carbohydrate quality, measured by dietary GI and GL, with ovarian reserve assessed by AFC. Methods: This study included 653 females from the Environment And Reproductive Health Study who completed AFC and food frequency questionnaire. Of these, 579 female individuals had a quantifiable AFC in both ovaries and were included in the primary analysis. We estimated average GI and GL for each participant from self-reported intakes of carbohydrate-containing foods and divided participants into tertiles. Poisson regression models were used to quantify the relations of GI, GL, carbohydrates, and AFC while adjusting for potential confounders. Results: Participants had a median age of 35 y. Compared to participants in the lowest tertile of dietary GI, those in the highest tertile had a 6.3% (0.6%, 12.3%) higher AFC (*p*, trend 0.03) after adjustment for potential confounders. Stratified analyses revealed that the association between GI and AFC was present only among participants who had not undergone infertility evaluations. Conclusions: A higher dietary GI was associated with a higher AFC. Subgroup analyses among individuals who had not had a diagnostic evaluation of infertility before joining the study suggest that high-glycemic carbohydrates may be related to PCOM.

## 1. Introduction

Antral follicle count (AFC) is a well-validated ovarian reserve marker among reproductive-aged females [1]. Clinically, it can serve as a versatile tool to guide the diagnosis of diverse pathologies with ovarian involvement. A high follicle number per ovary (FNPO) is considered the most important ultrasound marker of polycystic ovarian morphology (PCOM) in diagnosing polycystic ovary syndrome (PCOS) [2]. Conversely, a low AFC may suggest diminished ovarian reserve (DOR) [3].

Modifiable factors, including some dietary factors and lifestyle factors, such as smoking, have been previously associated with ovarian reserve [4,5,6,7]. Although the amount and quality of dietary carbohydrates may be associated with ovarian function [8], evidence of their association with ovarian reserve is scarce. Glycemic index (GI), a measure of carbohydrate quality, represents the glycemic potential of carbohydrate-containing foods [9]. In contrast, glycemic load (GL) considers both the quality and quantity of carbohydrates consumed. High-GI or -GL diets have been associated with insulin resistance and a higher risk of developing type 2 diabetes, cardiovascular disease, and all-cause mortality [10,11]. Replacement of low-GI foods with high-GI foods reduces postprandial glucose response and aids treatment for individuals with insulin resistance or type 2 diabetes [12]. In terms of reproductive health, the potential benefits of consuming a diet lower in high-GI foods among patients with gestational diabetes or PCOS have been previously described [13,14]. Insulin resistance is a pivotal etiological component of PCOS [15], with 95% of obese women and 75% of lean women affected by this condition exhibiting insulin resistance [16]. Among patients with PCOS, an improvement in insulin resistance contributes not only to improved symptoms but is also associated with subsequent diseases like type 2 diabetes or cardiovascular disease. However, to our knowledge, no study has investigated the association of dietary GI and GL or total carbohydrate intake with AFC.

Therefore, this study aimed to investigate the associations of GI, GL, and carbohydrate intake with ovarian reserve measured as AFC. We hypothesized that a higher intake of carbohydrates and a higher dietary GI and GL would be associated with a higher AFC among females seeking fertility.

## 2. Materials and Methods

### 2.1. Study Population

Study participants were female individuals who participated in the Environmental And Reproductive Health (EARTH) Study, a prospective cohort of subfertile couples presenting to the Massachusetts General Hospital Fertility Center, which began in 2004 and finished enrollment in 2019 [17]. At the study baseline, participants completed a baseline questionnaire that included demographic, medical history, and lifestyle questions. Females aged 18–45 years at study enrollment were eligible for this study if they completed a food frequency questionnaire (FFQ) after April 2007, underwent AFC measurement, and did not use a hormonal treatment such as oral contraceptives or leuprorelin at AFC scan. Of the 877 female participants who joined the EARTH study between 2004 and 2019 (Appendix A), 653 fulfilled these criteria, and 579 participants who had quantitative AFC data were included in the main analysis (Appendix A). Excluded participants were less likely to be White and had a higher proportion of DOR as a primary infertility diagnosis and lower AFC. Furthermore, 84% of these participants were excluded due to not completing the FFQ, rather than AFC data availability. Dietary carbohydrate quality may be associated with not completing the FFQ [18]; however, not completing the FFQ is less likely to be related to AFC results. Therefore, we surmised that this difference would not lead to selection bias based on the exposure status. The human subject committees of the Massachusetts General Hospital and the Harvard T. H. Chan School of Public Health approved this study. All participants provided written informed consent at the enrollment of the study.

### 2.2. Dietary Assessment

Dietary intake was assessed by using a previously validated semi-quantitative FFQ in which participants reported how often, on average (never/almost never to ≥6 times per day), they consumed 131 food and beverage items as a commonly used unit or portion size in the past year [19,20,21,22,23]. In a validation study, the de-attenuated correlation for carbohydrate intake was 0.69, comparing FFQs to the average of two 7-day diet records and 0.73 comparing FFQs to the average of four 24 h recalls [21]. GI, which represents the impact of a specific food’s carbohydrate composition on two-hour postprandial glucose levels compared to white bread, was obtained from published databases [9,24,25]. GI is scored from 0 to 100, with pure glucose assigned a value of 100. Foods with a GI ≤ 55 are classified as low-GI, 56–69 as medium-GI, and ≥70 as high-GI. We calculated the average dietary GI for each participant by multiplying carbohydrate content per serving of each food item by the average number of servings of that food per day and by its GI value, summing these products and then dividing it by the participant’s total daily carbohydrate intake [26,27]. Because the amount of carbohydrates in an overall diet can vary, we also calculated GL by multiplying each food’s GI by the available carbohydrate content for each food item and the average daily amount of food consumption [26,28]. Then, these products were summed to calculate the total dietary GL for each participant. GI and GL were energy-adjusted using the residuals method. The residuals were calculated by the regression model with total caloric intake as the independent variable and absolute nutrient intake as the dependent variable [29].

### 2.3. Antral Follicle Count (AFC) Measurement

Reproductive endocrinology and infertility physicians at Massachusetts General Hospital Fertility Center assessed AFC, as part of infertility screening, using transvaginal ultrasound. This evaluation was typically conducted on the third day of the natural menstrual cycle. If participants did not have a natural cycle, it was performed on the third day of withdrawal bleeding following progesterone administration. Follicles with a diameter of 2–10 mm were counted at each ovary and then summed to calculate AFC [1].

### 2.4. Statistical Analysis

We first divided participants into tertiles based on their GI, GL, and total carbohydrate intake. We fit Poisson regression models (generalized linear models with Poisson distribution and log-link function) with age and daily calorie intake to evaluate the relationship between each exposure and AFC. Results are presented as the relative difference (%) in AFC, with the lowest intake tertile as the reference. We adjusted for potential confounding variables selected based on our previous scientific knowledge and where baseline variables had significantly different distributions across the tertiles. Continuous variables included age, BMI, physical activity, calorie intake, alcohol intake, and caffeine intake. We also adjusted for categorical variables: race and ethnicity (non-Hispanic White [reference], non-Hispanic Black, non-Hispanic Asian, non-Hispanic other, Hispanic), smoking status (never smoker [reference], ever smoker, missing), education status (higher than college graduation), and multivitamin supplement use (yes, no [reference], missing).

### 2.5. Sensitivity Analyses

We conducted sensitivity analyses in which all AFC greater than 30 were truncated at 30 to minimize the impact of high AFC values on the results. To account for potential changes in dietary habits as a result of gaining information over the course of undergoing diagnostic procedures, we conducted a sensitivity analysis by stratifying the analyses according to having had an infertility examination prior to study enrollment (yes/no).

Finally, to evaluate whether GI, GL, or total carbohydrate intake had different associations at either end of the AFC distribution, we dichotomized the AFC data and considered two other outcomes: PCOM defined as FNPO ≥ 20 in at least one ovary [30] and DOR defined as AFC < 7 following the Bologna criteria for a poor ovarian response [31]. Logistic regression models were fitted with each food intake tertile for the binary outcome of PCOM and DOR using the lowest tertile as the reference group. We also conducted the same models with the stratum of infertility screening examination status. All statistical analyses were conducted using SAS 9.4 (SAS Institute, Inc., Cary, NC, USA).

## 3. Results

The median (interquartile range [IQR]) age, BMI, and AFC were 35.0 (32.0–38.0) years, 23.4 (21.2–26.4) kg/m^2^, and 13 (9, 18), respectively (Table 1). The median (range) GI was 50.5 (30.9–60.2), the median GL was 100.9 (20.0–306.7), and the median total carbohydrate intake was 48.3 (16.9–70.0) percent of calories. Participants in the highest tertile of GI tended to be younger and more likely to be Black compared to those in the lowest tertile of GI. Participants with higher intake of carbohydrates had a higher BMI and a higher proportion of never-smokers (Appendix A). There was no significant difference in the primary infertility diagnosis based on dietary GI, GL, and total carbohydrate intake (Table 1, Appendix A). GI, GL, and total carbohydrate intake were modestly correlated with each other (Appendix A). Whole grains (16.9%), whole fruits (9.3%), and non-starchy vegetables (5.8%) accounted for one-third of total carbohydrate intake in this population (Appendix A). As a result, intake of high-quality carbohydrates as a proportion of total calories was higher among participants in this study than among reproductive-age female individuals in the general population (21.3% in EARTH versus 8.6% in NHANES) (Appendix A) [32]. Moreover, participants who had previously undergone a fertility evaluation at enrollment had significantly higher intake of whole fruits and significantly lower intake of potatoes than participants who had not (Appendix A).

Compared to participants in the lowest tertile of dietary GI, those in the middle and highest tertiles had a 1.6% (95% CI: −3.7%, 7.1%) and 6.2% (95% CI: 0.7%, 12.0%) higher AFC, respectively (*p*, trend 0.03) (Table 2). Adjustment for BMI and other potential confounders had a minimal impact on the association (Table 2). Conversely, higher carbohydrate intake was inversely associated with AFC. Participants in the highest tertile of total carbohydrate intake had a 7.7% (95% CI: 2.2%, 12.8%) lower AFC compared to participants in the lowest tertile of intake (Table 2). These associations were attenuated when AFC was truncated at 30 (Appendix A). GL was unrelated to AFC (Table 2). A similar pattern was observed in the analyses comparing the prevalence of PCOM in relation to GI, GL, and total carbohydrate intake (Table 3).

Having undergone an infertility evaluation before joining the study was associated with intake of healthy and unhealthy carbohydrates in a pattern that was suggestive of dietary changes in response to specific information gained during the diagnostic process. Hence, we examined these associations, stratified according to history of infertility examination. In these analyses, the positive association between GI and AFC was restricted to participants who had not had a previous infertility examination when enrolling in the study (Table 4). The multivariable-adjusted relative difference in AFC comparing participants in the highest tertile to participants in the lowest tertile of GI was 3.5% (95% CI: −2.6%, 10.1%) among participants with an infertility evaluation before joining the study and 21.1% (95% CI: 3.2%, 42.2%) among participants without history of infertility examination at baseline (Table 4). However, the inverse association of total carbohydrate intake with AFC was restricted to participants with previous infertility examinations (Table 4). These results were similar when AFC was truncated at 30 (Appendix A). The same pattern was observed when AFC data were dichotomized to compare the prevalence of PCOM according to GI, GL, and total carbohydrate intake (Appendix A).

Finally, there was no association of GI, GL, or carbohydrate intake with DOR in the entire cohort (Appendix A). In the analyses stratified by prior infertility examination status, there was a suggestion of a positive association of dietary GL and total carbohydrate intake with the prevalence of DOR among participants without a history of infertility examination (Appendix A). Nevertheless, these estimates might be imprecise because the number of cases was limited, with only approximately 20% of the participants not having undergone prior infertility examination.

## 4. Discussion

We evaluated the relation of total carbohydrate intake, dietary GI, and dietary GL with AFC among participants attending a fertility center. In agreement with our hypothesis, we found that a higher GI was associated with a higher AFC. However, contrary to our hypothesis, higher total carbohydrate intake was associated with a lower AFC. In analyses stratified according to history of infertility evaluation, which we conducted to address the possibility of reverse causation, the positive association with GI was observed only among participants who had not had a previous infertility evaluation at the time that they joined the study. On the other hand, the inverse relation of total carbohydrate intake with AFC was restricted to participants who had previously undergone an infertility evaluation. This association pattern is suggestive of the presence of reverse causation, whereby the inverse relation with total carbohydrate intake may be the result of dietary changes after undergoing diagnostic testing for infertility, while the positive association with GI may be reflective of a true biological relationship between carbohydrate quality and AFC. Given the uncertainty inherent in these findings, it is difficult to assess their clinical significance.

Cross-sectional studies in general, and particularly those evaluating volitionally modifiable factors, are known to be susceptible to reverse causation—that is, of finding associations that reflect behavioral changes made in response to health conditions. We believe that this phenomenon is at play in our study and its results should, thus, be interpreted with caution. Reverse causation has been observed in cross-sectional or case–control studies with retrospective diet assessment when subjects change their dietary behaviors in response to the diagnosis of medical conditions [33,34,35,36]. It is widely understood that lifestyle factors play a vital role in reproductive health [37], and there is a growing body of knowledge on the relationship between carbohydrate intake and fertility specifically [38,39,40]. It has been reported that patients presenting to fertility centers strongly seek information that may improve their fertility and partake in following a healthy lifestyle in the hope of improving their fertility treatment results [41,42]. Fertility patients are also more motivated to change their lifestyles than other patients [43].

In this study, we observed multiple suggestions that reverse causation may have been a result of changes in lifestyle. First, the study population as a whole had higher intake of high-quality carbohydrates compared to the general population in the US [32]. Second, participants who had undergone an infertility evaluation prior to joining the study—and hence presumably had more opportunities to make lifestyle changes in response to diagnostic information—had higher fruit intake and lower potato intake compared to participants who had not had an infertility examination at enrollment. Third, we saw diverging association patterns in the analyses stratified according to history of infertility examination at enrollment, whereby the total carbohydrate intake (in which healthy carbohydrates were overrepresented in this population) appeared to be protective regarding PCOM only among participants who had previously had an infertility evaluation. Together, these suggest that the overall findings may represent a combination of associations arising from reverse causation and associations that may reflect true underlying biological relations. In this situation, stratified analyses may be able to separate these two types of relationships. In this study, our stratified analyses restricted to participants who had not previously had an infertility evaluation when joining the study may provide insights into how dietary carbohydrates impact ovarian reserve.

To our knowledge, no prior study has investigated the association of dietary GI, GL, and total carbohydrate intake with AFC. However, our findings are consistent with previous papers investigating the associations of the amount and quality of dietary carbohydrates with AMH or PCOS. Anderson et al. reported that the GL was positively associated with AMH concentration (β per 5 units = 0.051 [95% CI 0.008, 0.094]; *p*-trend = 0.020), while GI was not strongly related to AMH, among late premenopausal individuals from a prospective study cohort [4,44]. A case–control study revealed that participants with PCOS had a higher dietary GI than healthy controls [45]. In another case–control study, a higher GI, GL, and refined grain intake were positively associated with the risk of PCOS, while lower whole grain intake was inversely associated with the risk of PCOS [46]. Our findings are also consistent with studies focused on the management of PCOS. A meta-analysis of randomized controlled trials of PCOS patients found that low-GI diets improved glucoregulatory outcomes (HOMA-IR, insulin), lipid profiles, abdominal adiposity, and androgen status [47]. Additionally, a randomized controlled trial testing the effect of different dietary interventions in combination with physical activity found that PCOS patients assigned to low-GI or -GL diets had a significant reduction in FNPO after a 16-week intervention [48].

The positive associations of dietary GI with AFC and PCOM are biologically plausible. High-GI meals may contribute to insulin resistance and hyperglycemia [49]. PCOM is one of key features of PCOS that reflects the severity of reproductive dysfunction, such as androgen excess, obesity, and insulin resistance [50]. In this study, the model in which AFC greater than 30 were censored at 30 demonstrated an attenuated association of GI with AFC, suggesting that this relationship may be influenced more by participants with extensively high AFC, who might be at a high risk of PCOS. Mouse models revealed that glucose concentration affected the activation of mouse primordial follicles both in vitro and in vivo through the adenosine monophosphate-activated protein kinase/mammalian target of rapamycin signaling pathway, which may dysregulate the dynamics of ovarian reserve and/or impair the survival and competence of oocytes [51,52].

It is important to acknowledge the potential limitations of this study. First, given the cross-sectional design between exposure (FFQ) assessment and outcome (AFC) assessment, we need to consider the possibility of reverse causation, as previously discussed. However, by incorporating subgroup analyses that included different stages of infertility examination and treatment, we refined the interpretation of the results. Furthermore, as we used dietary data, which were collected using FFQs, there might have been mismeasurement. However, the FFQ has been extensively validated, including short-term dietary recalls and urinary and plasma concentration biomarkers, in previous studies [19,20,21,22,23]. Second, the study participants had a higher proportion of healthier carbohydrate intake than reported by the general US population. This difference in overall carbohydrate quality may affect the generalizability of the findings as it relates to the role of total carbohydrate intake and GL on ovarian reserve. It is important to re-examine this question in populations whose carbohydrate intake more closely resembles that of the general population. Another issue that can affect the generalizability of our findings is that the participants were predominantly White with a high level of education. Next, we could not utilize AMH levels to validate AFC data in this study as only one-third of the participants had AMH data. These data might contain potential selection bias for AMH data holders, as the study period coincided with the introduction of AMH into clinical practice and AMH assay development [6]. However, AFC is a highly reproducible measure with minimal inter- and intra-cycle variability [1,53], and the 2023 PCOS guidelines recommend that serum AMH not be used as a single test for the diagnosis of PCOS [2].

The strengths of our study lie in the availability of comprehensive information on dietary intake and potential confounding factors, collected using validated instruments, coupled with high-quality clinical data collected under standardized protocols. Moreover, this study had a relatively large sample size compared to other studies that treated ovarian function as an outcome. Further, our study is novel in investigating the direct association of dietary GI, GL, and carbohydrates with ovarian reserve as assessed by AFC.

## 5. Conclusions

A higher GI, indicating lower dietary carbohydrate quality, exhibited a positive linear association with AFC among subfertile female participants presenting at an academic fertility center. This suggests that a higher GI may be associated with PCOM. Although this finding is in agreement with emerging literature in support of the hypothesis that high-glycemic carbohydrates may be involved in PCOM and PCOS, our findings should be interpreted with caution due to study limitations. Therefore, the clinical impact of these findings remains uncertain. Further research on the relationship between dietary carbohydrates and ovarian reserve in populations less susceptible to reverse causation and with carbohydrate intakes more typical of the general population could provide additional insights.

## Figures and Tables

**Table 1 nutrients-17-00382-t001:** Demographic and reproductive characteristics of study participants, overall and according to categories of glycemic index ^a,b^.

		Total	Tertile 1 (Lowest)	Tertile 2 (Middle)	Tertile 3 (Highest)
n		653	217	219	217
Glycemic index, median (range)		50.50 (30.92–60.20)	46.70 (30.92–48.96)	50.50 (48.97–51.98)	53.81 (51.99–60.20)
Demographic characteristics					
Age (y)		35.0 (32.0–38.0)	36.0 (32.0–39.0)	35.0 (32.0–38.0)	34.0 (31.0–37.5)
BMI (kg/m^2^)		23.4 (21.2–26.4)	23.6 (21.5–26.4)	23.1 (21.1–25.8)	23.3 (21.2–26.8)
Race and ethnicity					
Non-Hispanic white, n (%)		514 (78.7)	174 (80.2)	170 (77.6)	170 (78.3)
Non-Hispanic Black, n (%)		27 (4.1)	6 (2.8)	6 (2.7)	15 (6.9)
Non-Hispanic Asian, n (%)		66 (10.1)	15 (6.9)	27 (12.3)	24 (11.1)
Non-Hispanic Other, n (%)		13 (2.0)	6 (2.8)	5 (2.3)	2 (0.9)
Hispanic, any race, n (%)		32 (4.9)	16 (7.4)	11 (5.0)	5 (2.3)
Smoking status, never, n (%)		485 (74.3)	153 (70.5)	166 (75.8)	166 (76.5)
Education, higher than college graduation, n (%)		568 (87.0)	190 (87.6)	191 (87.2)	187 (86.2)
Physical activity (hr/week)		5.0 (2.5–9.5)	6.4 (2.5–10.4)	5.0 (2.5–9.7)	4.5 (1.7–8.7)
Multivitamin intake, n (%)		554 (84.8)	189 (87.1)	191 (87.2)	174 (80.2)
Total calorie intake (kcal/day)		1682.3 (1363.1–2058.5)	1657.8 (1290.5–2062.3)	1686.3 (1363.1–2056.7)	1698.6 (1400.8–2058.5)
Carbohydrates (energy density [%])		48.3 (43.0–53.5)	43.9 (38.7–49.0)	50.6 (46.4–55.4)	49.7 (44.7–54.8)
Protein (energy density [%])		16.5 (14.9–18.6)	17.5 (15.5–19.4)	16.4 (14.7–18.1)	16.1 (14.7–17.9)
Total fat (energy density [%])		33.0 (29.5–37.4)	35.7 (31.9–40.4)	31.9 (28.9–35.1)	32.3 (28.8–36.1)
Fiber (g/day)		19.9 (15.3–26.4)	21.1 (15.8–28.8)	20.8 (16.0–27.5)	18.0 (13.9–24.3)
Total sugar (g/day)		82.4 (62.2–109.5)	79.4 (57.7–104.3)	90.9 (66.5–119.3)	80.3 (63.9–105.3)
Alcohol (mg/day)		4.7 (1.4–12.5)	7.3 (2.0–13.4)	4.7 (1.2–12.3)	4.1 (1.2–11.2)
Caffeine (mg/day)		105.0 (44.7–171.7)	120.6 (52.6–241.9)	97.6 (25.0–145.5)	101.5 (38.2–148.2)
Glycemic load		100.9 (75.8–129.9)	83.0 (61.3–106.4)	107.2 (81.6–133.9)	113.2 (89.4–140.4)
Reproductive history					
Previous infertility examination, n (%)		524 (80.3)	183 (84.3)	174 (79.5)	167 (77.0)
Previous infertility treatment, n (%)		311 (47.6)	103 (47.5)	107 (48.9)	101 (46.5)
History of past pregnancy, n (%)		278 (42.6)	101 (46.5)	91 (41.6)	86 (39.6)
Primary infertility diagnosis					
Male factor		159 (24.5)	48 (22.3)	61 (27.9)	50 (23.2)
Female factor	DOR	59 (9.1)	22 (10.2)	20 (9.1)	17 (7.9)
	Endometriosis	24 (3.7)	8 (3.7)	9 (4.1)	7 (3.2)
	Ovulatory	76 (11.7)	25 (11.6)	22 (10.1)	29 (13.4)
	Tubal	33 (5.1)	9 (4.2)	13 (5.9)	11 (5.1)
	Uterine	11 (1.7)	8 (3.7)	2 (0.9)	1 (0.5)
Unexplained		288 (44.3)	95 (44.2)	92 (42.0)	101 (46.8)

^a^ BMI: body mass index, DOR: diminished ovarian reserve; ^b^ data are presented as median (interquartile range) for continuous variables or n (%) for categorical variables.

**Table 2 nutrients-17-00382-t002:** Multivariable-adjusted associations of glycemic index, glycemic load, and carbohydrate intake with antral follicle count among 579 women who underwent numeric AFC measurement ^a,b^.

	Index Range	n	AFC	AFC, Relative Difference in Mean (95% CI) (%)	
			Median (IQR)	Age + Calorie-Adjusted Model	MV Model
Glycemic index	30.92–48.96	195	12.0 (8.0, 17.0)	Reference	Reference
	48.97–51.98	194	13.0 (9.0. 18.0)	1.6 (−3.7, 7.1)	1.0 (−4.4, 6.8)
	51.99–60.20	190	13.0 (10.0, 19.0)	6.2 (0.7, 12.0)	6.3 (0.6, 12.3)
	*p*, trend			*p* = 0.03	*p* = 0.03
Glycemic load	19.97–83.68	188	13.0 (9.0, 18.0)	Reference	Reference
	83.72–118.06	200	12.0 (9.0, 18.0)	−1.3 (−6.8, 4.6)	−1.1 (−6.7, 4.9)
	118.55–306.67	191	13.0 (9.0, 18.0)	2.3 (−5.3, 10.5)	2.2 (−5.8, 10.8)
	*p*, trend			*p* = 0.57	*p* = 0.60
Carbohydrate (energy density) (%)	16.58–44.68	191	13.0 (9.0, 18.0)	Reference	Reference
	44.70–51.67	194	13.0 (9.0, 18.0)	−4.3 (−9.3, 0.9)	−5.3 (−10.4, 0.0)
	51.69–69.98	194	12.0 (9.0, 19.0)	−5.3 (−10.2, −0.1)	−7.7 (−12.8, −2.2)
	*p*, trend			*p* = 0.05	*p* = 0.007

^a^ AFC: antral follicle count, IQR: interquartile range, MV: multivariable; ^b^ the Poisson regression models were adjusted for age, BMI, physical activity, calorie intake, alcohol intake, caffeine intake, race and ethnicity (non-Hispanic white [reference], non-Hispanic black, non-Hispanic Asian, non-Hispanic other, Hispanic), smoking status (never smoker [reference], ever smoker, missing), education status (higher than college graduation), and multivitamin supplement use (yes, no [reference], missing).

**Table 3 nutrients-17-00382-t003:** Multivariable-adjusted odds ratios of glycemic index, glycemic load, and carbohydrate with polycystic ovary morphology according to antral follicle counts among 651 participants ^a,b,c^.

	Index Range	n/Women (%)	Odds Ratio (95% CI) (%)	
			Age + Calorie	MV Model
Glycemic index	30.92–48.96	19/216 (8.8)	Reference	Reference
	48.97–51.98	18/218 (8.3)	0.82 (0.41, 1.63)	0.81 (0.40, 1.67)
	51.99–60.20	30/217 (13.8)	1.38 (0.74, 2.58)	1.40 (0.72, 2.71)
	*p*, trend		*p* = 0.29	*p* = 0.29
Glycemic load	19.97–83.68	26/216 (12.0)	Reference	Reference
	83.72–118.06	18/217 (8.3)	0.55 (0.28, 1.10)	0.54 (0.26, 1.10)
	118.55–306.67	23/218 (10.6)	0.55 (0.22, 1.36)	0.66 (0.25, 1.74)
	*p*, trend		*p* = 0.18	*p* = 0.36
Carbohydrate (energy density) (%)	16.58–44.68	25/217 (11.5)	Reference	Reference
	44.70–51.67	29/218 (13.3)	0.99 (0.55, 1.79)	0.98 (0.52, 1.83)
	51.69–69.98	13/216 (6.0)	0.40 (0.20, 0.83)	0.43 (0.20, 0.95)
	*p*, trend		*p* = 0.02	*p* = 0.05

^a^ MV: multivariable; ^b^ the logistic regression models were adjusted for age, BMI, physical activity, calorie intake, alcohol intake, caffeine intake, race and ethnicity (non-Hispanic white [reference], non-Hispanic black, non-Hispanic Asian, non-Hispanic other, Hispanic), smoking status (never smoker [reference], ever smoker, missing), education status (higher than college graduation), and multivitamin supplement use (yes, no [reference], missing); ^c^ polycystic ovarian morphology was defined as a follicle number per ovary ≥ 20 in at least one ovary.

**Table 4 nutrients-17-00382-t004:** Multivariable-adjusted associations of glycemic index, glycemic load, and carbohydrate intake with antral follicle count (AFC) among 579 women who underwent numeric AFC measurement according to previous infertility examination status ^a,b^.

Range	Women with History of Infertility Examination				Women without History of Infertility Examination				
	n	AFC Median (IQR)	Age + Calorie-Adjusted Model	MV Model	n	AFC Median (IQR)	Age + Calorie-Adjusted Model	MV Model	P_heterogeneity_ ^c^
Glycemic index									
30.92–48.96	165	12.0 (8.0, 17.0)	Reference	Reference	24	11.0 (8.0, 17.5)	Reference	Reference	0.11
48.97–51.98	154	13.0 (10.0, 18.0)	2.1 (−3.8, 8,3)	1.6 (−4.4, 8.0)	32	12.5 (9.0, 18.0)	−4.7 (−18.2, 11.2)	−3.2 (−18.4, 14.7)	
51.99–60.20	146	13.0 (9.0, 19.0)	4.4 (−1.6, 10.8)	3.5 (−2.6, 10.1)	37	13.0 (11.0, 20.0)	13.8 (−1.4, 31.4)	21.1 (3.2, 42.2)	
*p*, trend			*p* = 0.15	*p* = 0.27			*p* = 0.04	*p* = 0.006	
Glycemic load									
19.97–83.68	154	13.0 (9.0, 18.0)	Reference	Reference	28	14.5 (10.0, 19.0)	Reference	Reference	0.02
83.72–118.06	160	12.0 (9.0, 18.0)	1.2 (−5.0, 7.7)	1.8 (−4.6, 8.6)	36	12.5 (10.0, 18.5)	−12.5 (−24.3, 1.1)	−10,0 (−23.0, 5.3)	
118.55–306.67	151	13.0 (9.0, 18.0)	3.5 (−5.0, 12.6)	3.5 (−5.4, 13.2)	29	11.0 (9.0, 17.0)	−15.0 (−30.7, 4.3)	−8.5 (−27.0, 14.7)	
*p*, trend			*p* = 0.44	*p* = 0.45			*p* = 0.12	*p* = 0.44	
Carbohydrate (energy density) (%)									
16.58–44.68	153	13.0 (10.0, 18.0)	Reference	Reference	33	11.0 (8.0, 17.0)	Reference	Reference	0.20
44.70–51.67	145	13.0 (9.0, 18.0)	−6.6 (−12.0, −0.8)	−7.3 (−12.9, −1.4)	36	13.0 (9.0, 18.5)	−0.9 (−12.9, 12.8)	−3.9 (−16.4, 10.5)	
51.69–69.98	167	12.0 (9.0, 18.0)	−7.3 (−12.6, −1.8)	−10.0 (−15.5, −4.1)	24	15.5 (10.0, 19.0)	2.5 (−11.0, 18.2)	7.4 (−8,8, 26.4)	
*p*, trend			*p* = 0.01	*p* = 0.001			*p* = 0.75	*p* = 0.48	

^a^ AFC: antral follicle count, IQR: interquartile range, MV: multivariable. ^b^ The Poisson regression models were adjusted for age, BMI, physical activity, calorie intake, alcohol intake, caffeine intake, race and ethnicity (non-Hispanic white [reference], non-Hispanic black, non-Hispanic Asian, non-Hispanic other, Hispanic), smoking status (never smoker [reference], ever smoker, missing), education status (higher than college graduation), and multivitamin supplement use (yes, no [reference], missing). ^c^ Heterogeneity by infertility status was evaluated by adding the interaction term for infertility status and exposure in the models.

## Data Availability

Data described in the manuscript, code book, and analytic code will be made available upon request pending. Data are not publicly available due to privacy.

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
