# Peer review of "Association of Glycemic Index, Glycemic Load, and Carbohydrate Intake with Antral Follicle Counts Among Subfertile Females"

_nutrients, 2025, doi:10.3390/nu17030382_

Round 1

Reviewer 1 Report

Comments and Suggestions for Authors

line 58-60: "We hypothesized that a higher intake of carbohydrates and higher dietary GI and GL would be associated with a higher AFC among females seeking fertility." probably lower GI and GL and higher AFC

line 119-121: "(non-Hispanic White [ref], non-Hispanic Black, non-Hispanic Asian, non-Hispanic other, Hispanic), smoking status (never smoker[ref], ever smoker, missing), education status (higher than college graduation), and multivitamin supplement use (yes, no [ref], missing). " What does [ref] mean?

line 217-218: " (non-Hispanic white[ref], non-Hispanic black, non-Hispanic Asian, non-Hispanic other, Hispanic), smoking status (Never smoker[ref], ever smoker, missing), education status (higher than college graduation), and multivitamin supplement use (yes, no[ref], " What does [ref] mean?

line 227-228: " (non-Hispanic white[ref], non-Hispanic black, non-Hispanic Asian, non-Hispanic other, Hispanic), smoking status (Never smoker[ref], ever smoker, missing), education status (higher than college graduation), and multivitamin supplement use (yes, no[ref], " What does [ref] mean?

line 235-237: "y (non-Hispanic white[ref], non-Hispanic black, non-Hispanic Asian, non-Hispanic other, Hispanic), smoking status (Never smoker[ref], ever smoker, missing), education status (higher than college graduation), and multivitamin supplement use (yes, no[ref]," What does [ref] mean?

line 435-437: " (non-Hispanic white[ref], non-Hispanic black, non-Hispanic Asian, non-Hispanic other, Hispanic), smoking status (Never smoker[ref], ever smoker, missing), education status (higher than college graduation), and multivitamin supplement use (yes, no[ref]," What does [ref] mean?

I don't understand what "Ref" means in the tables... maybe it's worth explaining it under the table?

Table 4 and 9: Carbohydrate (Energy density) - value add unit %

line 242 :"we found that GI was associated with higher AFC. " should be “we found that LOWER GI was associated with higher AFC”

line 242-243: "However, Contrary to our hypothesis, total carbohydrate intake was associated with lower AFC." meaning that if someone did not consume carbohydrates, they had higher AFC? I think the words lower/higher are missing when talking about carbohydrates.

Author Response

Comments 1: line 58-60: "We hypothesized that a higher intake of carbohydrates and higher dietary GI and GL would be associated with a higher AFC among females seeking fertility." probably lower GI and GL and higher AFC.

Response 1: Thank you for pointing this out. We respectfully disagree with the suggestion. Since a lower GI and GL diet has been reported as a treatment for PCOS, we hypothesized that a higher GI and GL would be associated with higher AFC, as a marker of PCOM, rather than as a marker of ovarian reserve.

Comments 2: line 119-121: "(non-Hispanic White [ref], non-Hispanic Black, non-Hispanic Asian, non-Hispanic other, Hispanic), smoking status (never smoker[ref], ever smoker, missing), education status (higher than college graduation), and multivitamin supplement use (yes, no [ref], missing). " What does [ref] mean?
line 217-218: " (non-Hispanic white[ref], non-Hispanic black, non-Hispanic Asian, nonHispanic other, Hispanic), smoking status (Never smoker[ref], ever smoker, missing), education status (higher than college graduation), and multivitamin supplement use (yes, no[ref], " What does [ref] mean?
line 227-228: " (non-Hispanic white[ref], non-Hispanic black, non-Hispanic Asian, nonHispanic other, Hispanic), smoking status (Never smoker[ref], ever smoker, missing),
education status (higher than college graduation), and multivitamin supplement use
(yes, no[ref], " What does [ref] mean?
line 235-237: "y (non-Hispanic white[ref], non-Hispanic black, non-Hispanic Asian, nonHispanic other, Hispanic), smoking status (Never smoker[ref], ever smoker, missing),
education status (higher than college graduation), and multivitamin supplement use
(yes, no[ref]," What does [ref] mean?
line 435-437: " (non-Hispanic white[ref], non-Hispanic black, non-Hispanic Asian, nonHispanic other, Hispanic), smoking status (Never smoker[ref], ever smoker, missing),
education status (higher than college graduation), and multivitamin supplement use
(yes, no[ref]," What does [ref] mean?
I don't understand what "Ref" means in the tables... maybe it's worth explaining it under the table?

Response 2: Thank you for your comments. We used 'ref' as an abbreviation for 'reference.' Accordingly, we have replaced 'ref' with 'reference' in the manuscript, as well as in all tables where 'ref' appeared in the cells.

Response 3: I added % in the tables accordingly.

Comments 4: line 242 :"we found that GI was associated with higher AFC. " should be “we found that LOWER GI was associated with higher AFC”

Response 4: Thank you for your suggestion. We modified the sentence to “we found that higher GI was associated with higher AFC.” – line 248

Comments 5: line 242-243: "However, Contrary to our hypothesis, total carbohydrate intake was associated with lower AFC." meaning that if someone did not consume

carbohydrates, they had higher AFC? I think the words lower/higher are missing

when talking about carbohydrates.

Response 5: We modified the sentence as below. "However, Contrary to our hypothesis, higher total carbohydrate intake was associated with lower AFC." – line 249

Reviewer 2 Report

Comments and Suggestions for Authors

This study appears to be the first to investigate the relationship between dietary glycemic index (GI), glycemic load (GL), and total carbohydrate intake with ovarian reserve measured by antral follicle count (AFC). This fills a gap as previous studies did not explore these dietary factors in connection to ovarian reserve. It provides detailed analyses of dietary impacts by considering subfertile populations attending a fertility center. The investigation into the correlation between high-GI diets and AFC is novel, particularly in highlighting possible biological mechanisms linking carbohydrate quality to ovarian reserve markers. Based on the document, here are the specific improvements regarding the methodology and further controls the authors should consider:

1)Expanding to a more diverse population in terms of socioeconomic status, race, and ethnicity would improve generalizability.

2)Food Frequency Questionnaires (FFQs) have limitations in accurately capturing dietary intake, especially long-term habits. Adding biomarkers or short-term dietary recalls could validate and complement FFQ data.

3)Incorporate comprehensive controls for exercise patterns, stress levels, and adherence to dietary changes following infertility evaluations, as these could influence ovarian reserve.

Addressing these points would significantly enhance the rigor and interpretability of the study's findings.

Author Response

Comments 1: Expanding to a more diverse population in terms of socioeconomic status, race, and ethnicity would improve generalizability.

Response 1: Thank you for your suggestion. We acknowledge the importance of including a more diverse population to enhance the generalizability of our findings and have already noted this in the limitation (line 333-334). Since we have already included all the available data, we will consider it in future research. However, there is no perfect dataset, and in the National Survey of Family Growth1, approximately 60-70% of the respondents are White. Therefore, we believe that our findings provide valuable insights for those in the U.S. who require infertility treatment.

Comments 2: Food Frequency Questionnaires (FFQs) have limitations in accurately capturing dietary intake, especially long-term habits. Adding biomarkers or short-term dietary recalls could validate and complement FFQ data.

Response 2: Thank you for your suggestion. As mentioned in our manuscript, the FFQ has been extensively validated including short-term dietary recalls and urinary and plasma concentration biomarkers in previous studies 2–6 (line 89-90). Also, we added this limitation of the FFQ in the limitation section (please see line 324-327). We hope this addresses your concern. 

Comments 3: Incorporate comprehensive controls for exercise patterns, stress levels, and adherence to dietary changes following infertility evaluations, as these could influence ovarian reserve.

Response 3: Thank you for your suggestions. We understand the importance of considering behavioral change after the study enrollment and fertility evaluation. However, dietary patterns were reported only at the baseline in the EARTH study. We assessed various sports activities to determine exercise patterns, categorizing them into total physical activity, moderate to vigorous physical activity, and vigorous physical activity. Phycological stress was evaluated using the Perceived Stress Scale 4 (PSS-4)7. To address the reviewer’s concern, we considered the impact of modeling physical activity as time spent on vigorous activities only with additional adjustment for psychological stress as assessed with the PSS-4. As the reviewer can see in the table below, point estimates and confidence intervals in the two sets of models are very close to each other. While some of the estimates no longer reach conventional levels of statistical significance, this is most likely due to the fact that the sample size was reduced by 70 participants due to data availability. Therefore, we decided to retain the main model as it is. (Please find the table in the attached Word file).

Reference:

1. Nugent CN, Chandra A. Infertility and Impaired Fecundity in Women and Men in the United States, 2015–2019. National Center for Health Statistics (U.S.); 2024. doi:10.15620/cdc/147886
2. Willett WC, Sampson LK, Stampfer MJ, et al. Reproducibility and Validity of a Semiquantitative Food Frequency Questionnaire 2017. Am J Epidemiol. 1985;185(11):1109-1123. doi:10.1093/aje/kwx107
3. Rimm EB, Giovannucci EL, Stampfer MJ, Colditz GA, Litin LB, Willett WC. Reproducibility and Validity of an Expanded Self-Administered Semiquantitative Food Frequency Questionnaire among Male Health Professionals. Am J Epidemiol. 1992;135(10):1114-1126. doi:10.1093/oxfordjournals.aje.a116211
4. Yuan C, Spiegelman D, Rimm EB, et al. Validity of a Dietary Questionnaire Assessed by Comparison With Multiple Weighed Dietary Records or 24-Hour Recalls. Am J Epidemiol. 2017;185(7):570-584. doi:10.1093/aje/kww104
5. Yuan C, Spiegelman D, Rimm EB, et al. Relative Validity of Nutrient Intakes Assessed by Questionnaire, 24-Hour Recalls, and Diet Records as Compared With Urinary Recovery and Plasma Concentration Biomarkers: Findings for Women. Am J Epidemiol. 2018;187(5):1051-1063. doi:10.1093/aje/kwx328
6. Al-Shaar L, Yuan C, Rosner B, et al. Reproducibility and Validity of a Semiquantitative Food Frequency Questionnaire in Men Assessed by Multiple Methods. Am J Epidemiol. 2020;190(6):1122-1132. doi:10.1093/aje/kwaa280
7. Warttig SL, Forshaw MJ, South J, White AK. New, normative, English-sample data for the Short Form Perceived Stress Scale (PSS-4). J Health Psychol. 2013;18(12):1617-1628. doi:10.1177/1359105313508346

Reviewer 3 Report

Comments and Suggestions for Authors

The manuscript by Mitsunami et al., "Association of glycemic index, glycemic load, and carbohydrate intake with antral follicle counts among subfertile females". While the effort put into this paper is appreciated, and the findings are valuable, the manuscript needs to be improved.

Remove * for the EARTH Study Team from the list of authors

Abstract

Define GI in the first line of the abstract. Define PCOM. The abstract is too long; try to synthesize the information better. 

Introduction

This section is too short. Authors should describe the importance of insulin resistance. Authors must also include the study's utility after the aim; what are the perspectives?  

Material and Methods

The study period is not defined. Did it start in 2004 or 2007? Why the Approval Date of December 13, 1996? How did you establish which are the subfertile women as declared in the title of the manuscprit?

It is not clear to me why you used progesterone. Does this not impact the study's results? You don't speak about this in the description of the Study population. This is the only place where you mention progesterone. Were the women subjected to hormonal treatments for menstrual cycle regulation? Or what are we supposed to understand? You declared in line 69 that they did not use hormonal treatment.

Line 106 what is  "anatural menstrual cycle" ?

Line 119, 121 [ref]???

The same ref appears in the legend of Table 2 and table 3.

Discussion

Lines 243-246 You declare a previous evaluation of infertility evaluation - is there a reference to cite?

Author Response

Comments 1: Remove * for the EARTH Study Team from the list of authors

Response 1: We would respectfully like to insist on adding The EARTH Study Team as a corporate author. We realize that in the initial submission we did not provide a list of the members of the EARTH Study Team, which we are including in this revised submission. We would like to point out that our group has published 86 peer-reviewed manuscripts emanating from this study that include this corporate authorship in various medical journals. We include the corporate author because for any one paper not all members of the team meet the criteria for individual authorship as per ICMJE guidelines but have nevertheless made significant contributions to the study without which each individual publication would not have been possible. This practice is aligned with guidelines for corporate authors by ICMJE. We therefore respectfully request keeping the EARTH Study Team as a listed corporate author adding a supplemental note including a list of members of the team. We have attached the Excel file containing a list of non-author collaborators as supplemental material for this revision. Unfortunately, we cannot attach it to this reply since only Word or PDF files are permitted.

Comments 2: Abstract, Define GI in the first line of the abstract. Define PCOM. Response 2: I modified these accordingly.

Comments 3: The abstract is too long; try to synthesize the information better.

Response 3: We have edited the abstract below.

“ Background/Objectives: Few studies have investigated the association of dietary glycemic index (GI), glycemic load (GL), and carbohydrate intake with antral follicle counts (AFC). This study aimed to investigate the association of total carbohydrate intake and carbohydrate quality, measured by dietary GI and GL with ovarian reserve assessed by AFC. Methods: This study in-cluded 653 females from the Environment And Reproductive Health Study who completed AFC assessment and a food frequency. Of these, 579 female individuals had a quantifiable AFC in both ovaries and were included in the primary analysis. We estimated average GI and GL for each participant from self-reported intakes of carbohydrate-containing foods and divided participants into tertiles. Poisson regression models were used to quantify the relation of GI, GL and carbo-hydrates with AFC while adjusting for potential confounders. Results: Participants had a median age of 35 y. Compared to participants in the lowest tertile of dietary GI, those in the highest tertile had a 6.3% (0.6%, 12.3%) higher AFC (p, trend 0.03) after adjustment for potential confounders. Stratified analyses revealed the association between GI and AFC was present only among par-ticipants who had not undergone infertility evaluations. Conclusions: Higher dietary GI was associated with higher AFC. Subgroup analyses among individuals who had not had a diagnostic evaluation of infertility before joining the study suggest that high glycemic carbohydrates may be related to PCOM.“

Comments 4: Introduction, This section is too short. Authors should describe the importance of insulin resistance. Authors must also include the study's utility after the aim; what are the perspectives?

Response 4: Thank you for your suggestions. We lengthened Introduction by mentioning insulin resistance as follows. The added sections are highlighted with underlines. -line49-51, line53-55

“High GI or GL diets have been associated with insulin resistance and a higher risk of developing type 2 diabetes, cardiovascular disease, and all-cause mortality [10], [11]. Replacement of low-GI foods with high-GI foods reduces postprandial glucose response and treatment for individuals with insulin resistance or type 2 diabetes [12] In terms of reproductive health, the potential benefits of consuming a diet lower in high GI foods among patients with gestational diabetes or PCOS have been previously described [13], [14]. Insulin resistance is a pivotal etiological component of PCOS [15], with 95% of obese women and 75% of lean women affected by this condition exhibiting insulin resistance [16]. Among patients with PCOS, improvement of insulin resistance contributes not only to improved symptoms but is also associated with subsequent diseases like type 2 diabetes or cardiovascular disease.

Comments 5: Material and Methods, The study period is not defined. Did it start in 2004 or 2007?

Response 5: We apologize for getting you confused. The EARTH study was initiated in 2004, whereas FFQ was introduced to the study in 2007. I added the definition of the study period. “which began in 2004 and finished enrollment in 2019 ” -line 68

Comments 6: Why the Approval Date of December 13, 1996?

Response 6: Thank you for your comment. The initial IRB approval date of December 13, 1996, refers to the original protocol, which was focused solely on the enrollment of males due to the study's initial focus on male factor infertility. In 1999, the IRB record system was updated, and as a result, all protocols approved before that year were assigned "1999" in their protocol names. In 2004, the protocol was modified with additional funding to expand the study to include the recruitment of females in couples undergoing infertility evaluation or treatment at the same center, as part of the EARTH study. Despite these changes, the protocol name remains #1999P008167.

Comments 7: How did you establish which are the subfertile women as declared in the title of the manuscript?

Response 7: The EARTH Study recruits women and men seeking fertility evaluation and medically assisted reproductive treatment at an academic fertility center in the US. The study benefits from strong support and collaboration with physicians and other medical personnel from the fertility center. They identified potentially eligible patients during their care and informed them about the study at any point, including the start of their fertility evaluation such as ultrasound, hormonal evaluation, and semen analysis, or after initiating treatment. We then collected clinical information prospectively.

Comments 8: It is not clear to me why you used progesterone. Does this not impact the study's results? You don't speak about this in the description of the Study population. This is the only place where you mention progesterone. Were the women subjected to hormonal treatments for menstrual cycle regulation? Or what are we supposed to understand? You declared in line 69 that they did not use hormonal treatment.

Response 8: Thank you for giving us an opportunity to explain it. When evaluating AFC, assessments are typically conducted on the third day of a natural cycle. However, if menstruation does not occur naturally, progesterone is administered to induce withdrawal bleeding, and AFC is then evaluated on the third day of the withdrawal bleeding. It has been reported in the past that short-term progesterone administration does not affect AFC numbers1. The hormonal treatments mentioned in Line 69 refer to continuous use of oral contraceptives, hormone replacement therapy, or leuprorelin, which can affect AFC, and do not include progesterone used for a single cycle to induce withdrawal bleeding. While this might be a common scenario among reproductive endocrinologists, we have included an explanation for the benefit of readers of Nutrients.

Females aged 18-45 years at study enrollment were eligible for this study if they completed a food frequency questionnaire (FFQ) after April 2007, underwent AFC measurement, and did not use a hormonal treatment such as oral contraceptive or leuprorelin at the AFC scan.”-line 73

“This evaluation was typically conducted on the third day of the natural menstrual cycle. If participants did not have a natural cycle, it was performed on the third day of withdrawal bleeding following progesterone administration.” -line 110-111

Comments 9: Line 106 what is  "anatural menstrual cycle" ?

Response 9: We apologize for typo. That is “natural menstrual cycle”.

Comments 10: Line 119, 121 [ref]??? The same ref appears in the legend of Table 2 and table 3.

Response 10: Thank you for your comments. We used 'ref' as an abbreviation for 'reference.' Accordingly, we have replaced 'ref' with 'reference' in the manuscript, as well as in all tables where 'ref' appeared in the cells.

Comments 11: Discussion, Lines 243-246 You declare a previous evaluation of infertility evaluation - is there a reference to cite?

Response 11: Here, since we explained our findings based on the participants’ status of the infertility evaluation, there was nothing to cite.

Reference:

  1. Cédrin-Durnerin I, Bständig B, Parneix I, et al. Effects of oral contraceptive, synthetic progestogen or natural estrogen pre-treatments on the hormonal profile and the antral follicle cohort before GnRH antagonist protocol. Hum Reprod. 2007;22(1):109-116. doi:10.1093/humrep/del340

Round 2

Reviewer 3 Report

Comments and Suggestions for Authors

The authors responded to all the issues raised in the previous review.